# Whole-Exome Sequencing in a Family with an Unexplained Tendency for Venous Thromboembolism: Multicomponent Prediction of Low-Frequency Variant Deleteriousness and of Individual Protein Interaction

**DOI:** 10.3390/ijms241813809

**Published:** 2023-09-07

**Authors:** Barbara Lunghi, Nicole Ziliotto, Dario Balestra, Lucrezia Rossi, Patrizia Della Valle, Pasquale Pignatelli, Mirko Pinotti, Armando D’Angelo, Giovanna Marchetti, Francesco Bernardi

**Affiliations:** 1Department of Life Sciences and Biotechnology, University of Ferrara, 44121 Ferrara, Italy; lngbbr@unife.it (B.L.); blsdra@unife.it (D.B.); lucrezia.rossi@unife.it (L.R.); pnm@unife.it (M.P.); 2Department of Pharmacy, University of Pisa, 56126 Pisa, Italy; nicole.ziliotto@unipi.it; 3Unit of Coagulation Service and Thrombosis Research, IRCCS San Raffaele Hospital, 20132 Milan, Italy; dellavalle.patrizia@hsr.it (P.D.V.); dangelo.armando@hsr.it (A.D.); 4Department of Clinical Internal, Anesthesiological, and Cardiovascular Sciences, Sapienza University of Rome, 00185 Rome, Italy; pasquale.pignatelli@uniroma1.it; 5Department of Neuroscience and Rehabilitation, University of Ferrara, 44121 Ferrara, Italy; mrg@unife.it

**Keywords:** venous thromboembolism, WES analysis, family studies, low-frequency variants, protein interaction pattern, *CRP*, *F2*, *SERPINA1*, *THBS1*, *VWF*

## Abstract

Whole-exome sequencing (WES) in families with an unexplained tendency for venous thromboembolism (VTE) may favor detection of low-frequency variants in genes with known contribution to hemostasis or associated with VTE-related phenotypes. WES analysis in six family members, three of whom affected by documented VTE, filtered for MAF < 0.04 in 192 candidate genes, revealed 22 heterozygous (16 missense and six synonymous) variants in patients. Functional prediction by multi-component bioinformatics tools, implemented by a database/literature search, including ClinVar annotation and QTL analysis, prioritized 12 missense variants, three of which (*CRP* Leu61Pro, *F2* Asn514Lys and *NQO1* Arg139Trp) were present in all patients, and the frequent functional variants *FGB* Arg478Lys and *IL1A* Ala114Ser. Combinations of prioritized variants in each patient were used to infer functional protein interactions. Different interaction patterns, supported by high-quality evidence, included eight proteins intertwined in the “acute phase” (CRP, F2, SERPINA1 and IL1A) and/or in the “fibrinogen complex” (CRP, F2, PLAT, THBS1, VWF and FGB) significantly enriched terms. In a wide group of candidate genes, this approach highlighted six low-frequency variants (*CRP* Leu61Pro, *F2* Asn514Lys, *SERPINA1* Arg63Cys, *THBS1* Asp901Glu, *VWF* Arg1399His and *PLAT* Arg164Trp), five of which were top ranked for predicted deleteriousness, which in different combinations may contribute to disease susceptibility in members of this family.

## 1. Introduction

Venous thromboembolism (VTE) is a complex multifactorial disorder [1,2], in which the genetic component is estimated to have a major role [3,4,5]. Historically, susceptibility genes for VTE, mainly codifying for protein of coagulation cascade and its control, were identified in family and association studies [6,7,8,9,10,11]. Many “private” variants in the anticoagulation genes were found [12], which when combined with each other and/or with specific environmental or lifestyle components greatly increase the risk of developing disease [13,14]. However, the known thrombophilia variants account for a minor fraction of VTE heritability [15].

In genome-wide association studies (GWAS) the associations for part of the candidate VTE genes were successfully replicated and new susceptibility genes were suggested [16,17,18,19], although for several variants the biologic impact appear of uncertain significance. Overall, a large proportion of VTE genetic components still remain unexplained.

Part of the “missing” heritability might be due to rare variants [20]. These variants, not captured by a GWAS approach or by genotype imputation but detectable by high throughput sequencing technologies [21] might have even a larger genetic effect than common variants for VTE etiology and could be of clinical importance [21]. Indeed, the whole-exome sequencing (WES) approach in the evaluation of patients with VTE [22,23,24,25] has provided multiple novel genetic variants with predicted roles in thrombosis or thrombophilia, thus extending the panel of candidate genes. 

More recently, through genomic-transcriptomic-wide analysis [26], single and multimarker genetic testing [27] and large meta-analyzed GWAS [28], novel genetic risk modifiers for VTE have been suggested to contribute, even with small effects, to VTE susceptibility. Many of the recently identified loci, being outside of known or currently hypothesized pathways for thrombosis, suggest new molecular components belonging to platelet and blood biology, inflammation and immuno-mediated processes, potentially contributing to VTE susceptibility [29,30]. 

Whether WES could contribute to molecular diagnosis has been investigated in a few families with unexplained VTE and no recognized thrombophilic defects. Novel rare variants responsible for inherited thrombophilia, in the prothrombin gene [31] or outside the coagulation cascade [32,33] have been reported. Some rare variants in genes, not previously reported to be associated with VTE, and likely with an impact on the risk of VTE, were identified in two large pedigrees [34], although their relevance as novel thrombophilic defects was not confirmed. 

Prompted by the small number of family studies and by the extended panel of candidate genes, WES analysis was conducted in a family with three subjects experiencing documented VTE, without being carriers of recognized thrombophilic gene variants or of anticoagulant protein deficiencies. WES analysis, combined with multiple bioinformatics approaches and public database mining, was focused on low-frequency missense variants in 192 candidate genes that have been previously suggested for their role in VTE or associated with VTE-related phenotypes.

## 2. Results

A schematic flow chart of methodology and data analysis is shown in Figure 1.

The family under study (Appendix A) was selected for genetic analysis by WES based on the documented venous thromboembolism in three family members, negative routine thrombophilia testing and absence of conditions (smoking, diabetes, obesity and sedentary lifestyle) that might have favored VTE.

### 2.1. WES Analysis

The reference list of genes (n = 192, Appendix A) for the present study was generated by using as primary resource the PubMed database, for which the search terms “VTE genes” and “VTE GWAS” were inserted. 

The WES analysis in the wide panel of candidate genes, performed on six family members (Appendix A), three affected (II2, II3, III2) and three unaffected (I1, II1, III1), did not reveal common thrombophilic variants or point mutations affecting main coagulation inhibitor genes (SERPINC1, PROS, PROC). 

Appendix A reports missense changes with MAF 0.04–0.30 in the 1000 Genomes Project (1000G), their zygosity in patients and ClinVar annotation. Among 20 variants present in all affected subjects, the FGB Arg478Lys caused, after recombinant expression, higher clot stiffness and slower fibrinolysis rate [35], was associated with fibrinogen plasma levels and eQTLs, and displayed interaction with the relative amounts of γ’ fibrinogen [36]. However, the conferred DVT/PE risk was negligible [37]. The multiple in linkage F5 SNPs have been associated to decreased VTE risk (OR 0.77, 0.68–0.87) [38]. The LMOD1 rs2820312, in the homozygous condition in all affected subjects, belongs to novel secondary signals in previously established GWAS loci associated with BMI [39]. 

Among variants present in the propositus (III2), the KNG1 Ile197Met has been associated to decreased kininogen [40] and factor XI [41] levels, and the homozygous IL1A Ala114Ser to ~50% decreased IL-1α release, and in turn with cardiovascular disease [42]. 

Focusing on low-frequency variants, a MAF < 0.04 was selected as threshold for filtering through the Genome Aggregation (gnomAD3) and 1000 Genomes Project (1000G) Databases. Filtering revealed 22 variants (16 missense and six synonymous) heterozygous in at least one of the affected family members, six with MAF < 0.001 (Table 1). 

Twelve out of sixteen missense variants were carried by the propositus’ uncle (II3), and nine by the propositus (III2) and his father (II2). All affected family members (II2, II3, III2) carried (Table 1) five missense variants (CRP Leu61Pro, F2 Asn514Lys, JAK2 Leu393Val, NQO1 Arg139Trp and the new PSG8 Cys9Ser). High frequency of the variants was excluded in other populations (Appendix A).

Several low and high frequency variants were found in linkage or compound heterozygous condition. The PSG8 Cys9Ser, a newly detected missense variant that may influence signal peptide recognition (http://www.signalpeptide.de/index.php (accessed on 28 April 2023)), was heterozygous in all affected subjects, and compound heterozygous (III2) with PSG8 Gly86Ser and Ile88Arg (Appendix A). The THBS1 Asp901Glu, the other newly detected missense variant (II2 and III2), was linked to Thr523Ala and Asn700Ser (Appendix A). The SERPINA1 Arg63Cys (II2, III2) was found in linkage with Arg125His and Glu400Asp (Appendix A). The VWF Arg1399His and Asp1472His variants were in the compound heterozygosity (II3). 

### 2.2. Observed/Predicted Functional Impact of Variants

Six low-frequency variants were clinically annotated in the ClinVar, a public archive of reports of the relationships among human variations and phenotypes, with supporting evidence. Pathological phenotypes (LP and P) were reported for SERPINA1 Arg63Cys and for VWF Arg1399His, the latter with discrepant pathogenicity assessments (Table 1). Benign or likely benign phenotypes were annotated for JAK2 Leu393Val, and for three synonymous changes (MPL Ser414Ser, PLGC2 Phe382Phe and TMCO1 Leu162Leu).

The functional impact on mRNA expression (transcription and splicing processes) was investigated (Table 2). The ARID4A rs146509016 and NQO1 rs1131341 were located within a splice site region, and the NQO1 rs1131341 also in an open chromatin region. Enhancer sequences overlapped the exons containing the MPL rs544064034 and SERPIND1 rs35646566, and a binding site for the transcription factor CTCF encompassed the exon containing the PLGC2 rs138637229. 

The disruption of a 5′ splice site was predicted for the NQO1 variant, and of an exonic splicing enhancer (ESE) for eight variants. The creation of a new exonic splicing silencer (ESS) was predicted for four variants and of new 5′ splice sites for two (Table 2).

The quantitative trait locus (QTL) analysis (Table 2, GTEx portal) supported a significant functional impact of PLAT rs2020921 on mRNA levels and of NQO1 rs1131341 on splicing [43], and on mRNA level/splicing of other genes (Table 2) encoding several proteins: (i) IGLON5, a member of immunoglobulin superfamily IgLON; (ii) NOB1, a nuclease involved in rRNA processing; (iii) COG4, a protein of the conserved oligomeric Golgi complex; (iv) AP3M2, an AP-3 complex component with a role in protein trafficking to lysosomes and specialized organelles; (v) POLB, the polymerase beta and vi) SLC20A2, a phosphate transporter. The PEAR1 rs77795865 influences mRNA levels of LRRC71, found associated with platelet phenotypes in GWAS catalog. 

Concerning variants causing synonymous changes (Table 2), the PTGIS rs61322884 influences mRNA level (eQTL) of SLC9A8, which encodes a Golgi sodium–hydrogen exchanger and has been associated with chronic inflammatory [44] and coronary artery [45] diseases, in turn potentially related to VTE mechanisms. For the other synonymous variants, analysis provided hints for their involvement in regulatory elements of splicing (ESE and ESS) and of transcription (Table 2), albeit not reflected on recognized QTL. 

### 2.3. Predicted Impact of Variants on Protein Structure and Function

The potential impact of variants on protein structure and function and their pathogenicity were predicted [46] by using 15 multi-component bioinformatics tools, six of which (ClinPred, DANN Coding, MetaSVM, REVEL, VEST4 and FATHMM XF) were based on artificial intelligence. A profile of damaging effects, functional and disease associated, was predicted for each missense variant, referred to the main transcript (Appendix A). The degree of predicted deleteriousness is schematically represented in the heat maps (Figure 2) obtained from probability scores (left panel) or from prediction scores, categorized as neutral, moderate or damaging (right panel). The probability scores of REVEL, based on a combination of scores from 13 individual tools, were used to rank variants. Variants detected in at least two thrombotic subjects, classified as damaging by at least four bioinformatics tools, and those associated with eQTL/sQTL and annotated in ClinVar, were highlighted in Figure 2. No evidence for altered post-translational modification (acetylation, phosphorylation or ubiquitination) was found (Phosphosite Plus, https://www.phosphosite.org/homeAction (accessed on 28 April 2023)).

Among variants ranking from the ninth to the fifteenth position, those predicted as damaging by only two tools and without additional functional correlates, were not prioritized (UGT1A3, JAK2, SAA2 and PSG8, Figure 2). Overall, 12 missense variants were prioritized, three of which (CRP Leu61Pro, F2 Asn514Lys and NQO1 Arg139Trp) were present in all affected family members.

Among prioritized variants, KLK13 His109Tyr and NQO1 Arg139Trp ranked in the last quartile of predicted deleteriousness, but were characterized by QTL associations (Table 2) with body mass index and immune system through IGLON5 [47] and protein transport and glycosylation through COG4 [48]. 

Plasma assays specifically designed for the protein variants might support the predicted functional impact of prioritized missense changes. Global functional assays, thrombin generation and thromboelastometry, which may partially surrogate the protein specific investigation, were performed. In thrombin generation assays induced by low tissue factor concentration, of particular interest for the F2 Asn514Lys variant present in all patients, the extrinsic thrombin potential values in the propositus (1.01 nM/min) and his father (0.9 nM/min) were indistinguishable from normal range (0.88–1.12 nM/min). In the thromboelastometry experiments, to detect “in vitro” hypo-fibrinolysis conditions of interest for VTE, the values of clotting time, maximum clot firmness, lysis onset time and lysis time in patient II3, heterozygous for the PLAT Arg164Trp variant, did not differ from normal.

### 2.4. Interaction among Proteins Containing Variants in the Affected Family Members

Virtually all substitutions were predicted by modelling to be located on protein surface, were not expected to cause major structural damage, and could perturb domain–domain and protein–protein interactions.

Functional association of proteins containing prioritized low-frequency missense variants were explored in the STRING database. In addition, the frequent and functional variants FGB Arg478Lys, present in all affected family members, and IL1A Ala114Ser, homozygous only in the propositus (Appendix A), were also prioritized. Accordingly, Fibrinogen Beta (FGB) and interleukin 1A (IL1A) were included in the interaction analysis. Taking into account the different combinations of variants in the affected subjects, and that several variants with the highest ranking of predicted deleteriousness were not present in all affected subjects, we explored individual protein combinations and interactions in the affected family members (Figure 3). 

The affected family members were characterized by five/six proteins’ interactions that remarkably differed in the propositus III2 (THBS1, F2, SERPINA1, CRP, FGB and IL1A) and his father II2 (THBS1, F2, SERPINA1, CRP and FGB) as compared with paternal uncle II3 (F2, CRP, VWF, PLAT and FGB). The predicted interactions were characterized for high (FGB with THBS1/SERPINA1/VWF/CRP, F2-SERPINA1, CRP with VWF/IL1A) and very high confidence (F2 with CRP/FGB/VWF and PLAT-VWF), supported by different types of evidence (Figure 3). 

Functional enrichment analysis showed that, among Biological Processes (Gene Ontology) the “acute phase response” (GO term:0006953), among Annotated Keywords (Uniprot) the “acute phase” (KW-0011) and among Compartments, the “fibrinogen complex” (GOCC:0005577), displayed highly significant enrichment. In subjects III2 and II2, the interaction pattern exhibited a balanced contribution of proteins in the “acute phase” and “fibrinogen complex”, the latter prevailing in the protein interaction pattern of subject II3.

## 3. Discussion

A family history of VTE indicates important underlying genetic components. In a family with multiple members experiencing VTE without being carriers of recognized thrombophilic defects, WES analysis was focused on low-frequency variants in multiple candidate genes. In relation to biological links to VTE, the genes carrying variants in the affected family members were grouped for belonging mainly to platelet, blood coagulation/fibrinolysis and inflammation processes (Figure 4). 

In WES studies, conducted with different filtering/prioritizing strategies in families with an unexplained tendency for VTE (Table 3), nonsense changes/deletions/low-frequency-SNPs homozygosity and genetic conditions supporting contribution to diseases were sporadically detected, as well as new variants. Since missense variants were the main output of all family studies (Table 3), their prioritization is crucial to select those that could play a role in VTE etiology. 

The SIFT tool, based on protein sequence homology and physical properties of amino-acids, and PhD-SNPg, a binary classifier trained and tested using >30,000 ClinVar pathogenic and benign SNVs, predicted deleteriousness for the vast majority of variants. Differently, most of the artificial intelligence-based tools, except FATHMM XF, predicted deleteriousness for only some of them. By implementing database/literature observations, functional correlates stemming from QTL analysis, prediction of splicing process involvement, and ClinVar annotation and 12 low-frequency variants, three variants present in all affected family members were prioritized together with two functional frequent variants. Since variants highest ranked for predicted deleteriousness were not present in all affected subjects, who remarkably differed in variant combination, potential protein–protein interaction were investigated in each affected family member (Figure 3). The different protein interaction patterns, supported by several pieces of high-quality evidence, included proteins intertwined in the “acute phase” (CRP, F2, SERPINA1 and IL1A) and/or in the “fibrinogen complex” (CRP, F2, PLAT, THBS1, VWF and FGB). Interestingly, five proteins carried missense variants top ranked for predicted deleteriousness (CRP Leu61Pro, F2 Asn514Lys, SERPINA1 Arg63Cys, THBS1 Asp901Glu and VWF Arg1399His). For the SERPINA1 Arg63Cys and VWF Arg1399His variants, pathogenicity assessment was also annotated in ClinVar [49,50]. It is worth noting that VWF Arg1399His has been found to be strongly associated with deep-vein thrombosis (OR 3.26, 95% CI 1.18–8.98) [50], at variance with most of the VWF missense changes, which predict bleeding conditions. Moreover, the THBS1 Asp901Glu, a newly detected variant, and the SERPINA1 Arg63Cys, associated with reduced inhibitory activity and increased polymerization susceptibility [49], were linked to missense variants with higher frequency thus resulting in proteins with multiple changes. 

We can tentatively speculate that partially different mutational backgrounds and remarkably different functional interactions predisposed the brothers II2 and II3 to VTE episodes at around 60 years of age for both of them. On the other hand, these subjects were both carriers of other prioritized variants that may be associated with platelet phenotypes either directly (*NQO1* and *PEAR1*, Figure 3) [51,52], or indirectly (*PEAR1*) through mRNA level of *LRRC71* [53]. In addition, the rare *F2* Asn514Lys variant is located in thrombin, a protein with a crucial role in platelet activation. Furthermore, it can be hypothesized that trauma, a strong acquired condition, could have substantially contributed to early onset of VTE in the propositus III2.

With an approach aimed at highlighting genetic components that may confer VTE susceptibility by interaction [14], six low-frequency variants (*CRP* Leu61Pro, *F2* Asn514Lys, *SERPINA1* Arg63Cys, *THBS1* Asp901Glu, *VWF* Arg1399His and *PLAT* Arg164Trp), the first two present in all affected family members, and the first five top ranked for deleteriousness, were the main findings of this family-based WES analysis (Table 3). The frequent functional variants *FGB* Arg478Lys and *IL1A* Ala114Ser have further boosted “acute phase” and “fibrinogen complex”-related interactions, which were highly significant also without fibrinogen beta and interleukin 1 alpha proteins inclusion. Taking into account that experimental investigation of protein variant combinations is still a very difficult task, this study has several limitations: (i) The low number of family members affected by VTE and the presence of only one elderly member not affected by VTE (I1). It is worth noting that the grandfather shares with the affected sons and nephew only three out of the six low frequency variants determining the protein interaction patterns; (ii) The contribution of synonymous variants was not further explored, despite their potential involvement in regulatory elements of splicing and of transcription. However, synonymous variant predictions were not reflected on recognized QTL; (iii) Exploring around 200 candidate genes favors prediction of interactions as new combinations of small size genetic effects. However, the low number of proteins with prioritized missense variants displayed multiple interactions with high confidence scores and identified statistically significant enriched terms; (iv) The functional implication of variants was not experimentally explored. It is worth noting that the amino acid substitutions affect exposed residues and do not suggest quantitative defects of proteins. Moreover, the variants were present in the heterozygous condition, which does not favor their functional investigation and (v) Although several variants were located in genes involved in platelet biology, the study design did not include platelet functional investigation. 

## 4. Materials and Methods

### 4.1. Clinical History of the Family and Laboratory Assays

Clinical history of the family members was obtained by the hospital discharge letters and by validated questionnaire [54]. The following criteria supported deep genetic analysis in the family: (i) documented pulmonary embolism episodes in the three family members associated with vein thrombosis in the three family members, which clearly define the phenotypes (see following description for each patient) under study; (ii) routine thrombophilia testing excluding the deficiencies of antithrombin, protein C, protein S, APC resistance, and high factor IX and factor VIII levels; (iii) absence of lupus anticoagulant, anti-cardiolipin-, anti-beta2GPI-(IgM, IgG) antibodies, extractable nuclear antigens and anti-nuclear antibodies and (iv) genetic assays excluding the factor V Leiden and prothrombin G20210A mutations. 

Routine assays revealed normal platelet number in the patients. The risks conferred by smoking, diabetes, obesity and sedentary lifestyle were excluded.

The propositus (III2, Appendix A) at the age of 18 developed deep-vein thrombosis (DVT) of the right popliteal vein after trauma of the ankle during sports activity. Twenty days after trauma, a CT scan after dyspnea revealed the presence of PE at the level of the pulmonary artery extended to the segmental branches.

After hospitalization for dyspnea at the age of 59, the propositus’ father (II2) was found to be affected by PE, confirmed by CT scan, and by superficial vein thrombosis at the small saphenous veins.

After hospitalization for thoracalgia at the age of 58, the propositus’s paternal uncle (II3) was found to be affected by acute PE confirmed by CT scan, and by femoral vein thrombosis. The patient reported recent pneumonia.

No additional family member has been affected by VTE episodes, as derived from validated questionnaire.

### 4.2. Global Functional Assays

Thrombin generation induced in plasma by 3 pM of tissue factor was performed as described [55]. Thromboelastometry (ROTEM) in plasma was performed as previously described to assay exogenous activation (recombinant tissue plasminogen activator) of fibrinolysis [56].

### 4.3. Selection of Candidates Genes

The primary source was the PubMed database, for which search terms “venous thromboembolism (VTE)” and “GWAS” were used, and a list of 192 genes was generated for the present study.

### 4.4. Whole-Exome Sequencing and Analysis

Genomic DNA was extracted from peripheral blood using the Wizard Genomic DNA Purification Kit (Promega, Madison, WI, USA). WES [57,58] was performed on six individuals, three diagnosed with VTE, from an Italian family by using the SureSelect Human Exon 6 exome capture kit (Agilent Technologies) and the NovaSeq platform (Illumina, San Diego, CA, USA) with 150-bp paired-end reads. Reads were mapped against the hg19 human reference sequence using SOAPaligner. Variants calling was performed by the Complete Genomics Small Variant Caller.

Genetic variations were verified in the database of Single-Nucleotide Polymorphisms (dbSNP, http://www.ncbi.mln.nih.gov/snp (accessed on 28 April 2023)) and their frequency was verified in the 1000 Genomes Project (1000G, all) and the Genome Aggregation database (gnomAD v3.1, all).

Base calling accuracy, measured by the Phred quality score (Q score), was 98.4% for Q > 20 and 95.5% for Q > 30. Filtering of variants was based on a quality score >30 and a minor allele frequency (MAF) < 0.04 in the three affected family member.

### 4.5. Analysis of the Functional Impact of Variants

The annotated variant’s impact on disease was verified in ClinVar (NCBI resource; accessed April 2023). The overlapped regulatory features were obtained from VEP, Variant Effect Predictor (Ensembl GRCh38 release 109, February 2023). The quantitative trait loci (QTL) analysis was obtained from GTEx portal (release V8, https://gtexportal.org/home/accessed (accessed on 28 April 2023)).

The functional impact of variants on splicing was investigated by the HOT-SKIP (https://hot-skip.img.cas.cz/ (accessed on 28 April 2023)) for the assessment of the impact on ESE and ESS, by the SpliceAI (https://spliceailookup.broadinstitute.org/# (accessed on 28 April 2023)) and SpliceRover (http://bioit2.irc.ugent.be/rover/splicerover/ (accessed on 28 April 2023)) for the impact on canonical sites. 

Information on gene function, and on SNPs—associated phenotypes were obtained from Gene Cards database (https://www.genecards.org (accessed on 28 April 2023)) and GWAS catalog (https://www.ebi.ac.uk/gwas/ (accessed on 28 April 2023)), accessed April 2023.

The potential impact of variants on secondary protein structure was explored by prediction of the structural changes introduced by an amino acid substitution, conducted by exploiting the Missense3D bioinformatic tool (http://missense3d.bc.ic.ac.uk/~missense3d/ (accessed on 28 April 2023)) [59].

The potential impact of variants on protein function and their pathogenicity were predicted by 15 algorithms. All analyses were automated by exploiting the OpenCRAVAT server (https://opencravat.org/ (accessed on 28 April 2023)) [46]. The algorithms are reported with a brief description: ClinPred is an efficient tool for identifying disease-relevant nonsynonymous variants. It is based on two machine learning algorithms that use existing pathogenicity scores and, notably, benefits from inclusion of normal population allele frequency from the gnomAD database as an input feature. DANNCoding is a deep learning approach for annotating the pathogenicity of genetic variants which uses the same feature set and training data as CADD to train a deep neural network (DNN). MetaSVM is an ensemble-based prediction algorithm developed by integrating 10 component scores (SIFT, PolyPhen-2 HDIV, PolyPhen-2 HVAR, GERP++, MutationTaster, Mutation Assessor, FATHMM, LRT, SiPhy, PhyloP) and the maximum frequency observed in the 1000 genomes populations, using a support vector machine model. REVEL is an ensemble method for predicting the pathogenicity of missense variants based on a combination of scores from 13 individual tools: MutPred, FATHMM v2.3, VEST 3.0, PolyPhen-2, SIFT, PROVEAN, MutationAssessor, MutationTaster, LRT, GERP++, SiPhy, phyloP and phastCons. VEST is a machine learning method that predicts the functional significance of missense mutations based on the probability that they are pathogenic. FATHMM-XF (FATHMM with eXtended Features) is an improvement of previous predictor, FATHMM-MKL, and predicts whether single nucleotide variants (SNVs) in the human genome are likely to be functional or non-functional in inherited diseases. MetaLR is an ensemble-based prediction algorithm developed by integrating 10 component scores (SIFT, PolyPhen-2 HDIV, PolyPhen-2 HVAR, GERP++, MutationTaster, Mutation Assessor, FATHMM, LRT, SiPhy and PhyloP) and the maximum frequency observed in the 1000 genomes populations, using a logistic regression model. PhD-SNPg is a binary classifier that implements Gradient Boosting-based algorithm for predicting pathogenic variants in coding and non-coding regions. Likelihood Ratio Test can accurately identify a subset of deleterious mutations that disrupt highly conserved amino acids within protein-coding sequences by using a comparative genomics data set of 32 vertebrate species. Mutation Assessor is a database providing prediction of the functional impact of amino-acid substitutions in proteins. Functional impact is calculated based on evolutionary conservation of the affected amino acid in protein homologs.

MutationTaster evaluates disease-causing potential of sequence alterations. PROVEAN (Protein Variant Effect Analyzer, Version 1.1.5) is a software tool which predicts whether an amino acid substitution or indel has an impact on the biological function of a protein. PolyPhen-2 (Polymorphism Phenotyping v2) is a tool which predicts possible impact of an amino acid substitution on the structure and function of a human protein using straightforward physical and comparative considerations. SIFT predicts whether an amino acid substitution affects protein function based on sequence homology and the physical properties of amino acids. 

Protein–protein interaction was predicted by STRING database (v.11.5, https://string-db.org/, accessed on June 2023) which collects and integrates protein–protein interactions, both physical interactions as well as functional associations. In the STRING setting the network edges were reported with the evidence mode, in which the types of evidence used in predicting the associations are shown as differently colored lines. Each protein–protein interaction is annotated with a score, which is indicator of confidence, the approximate probability that a predicted link exists between two proteins. Confidence scores are first computed separately per evidence type, and then integrated into a final, “combined” confidence score. Confidence limits are as follows: low confidence, 0.15; medium confidence, 0.4; high confidence, 0.7 and highest confidence, 0.9. 

In parallel with the network prediction, functional enrichment analysis was performed by STRING, which imports knowledge from the three Gene Ontology branches (Biological process, Molecular Function and Cellular Component), KEGG pathways, UniProtKB Keywords, COMPARTMENTS and TISSUES.

## 5. Conclusions

We believe that our approach has the potential to detect in wide groups of candidate genes and gene variations the most plausible combinations of small-size-effects variants supporting disease susceptibility in families with an unexplained tendency for VTE. Whether this strategy could be of clinical and diagnostic importance deserves further investigation in other families.

## Figures and Tables

**Figure 1 ijms-24-13809-f001:**
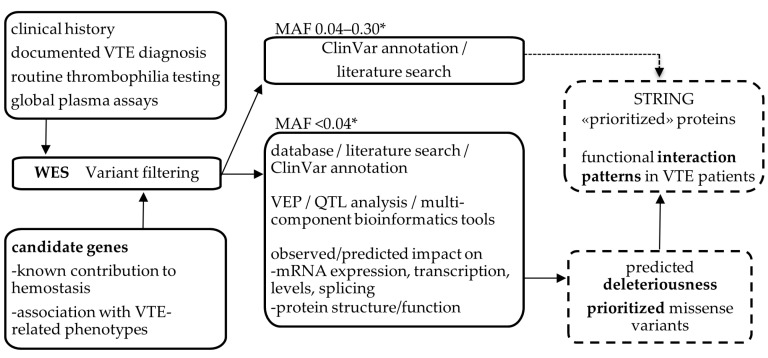
Overview of the family study providing essential information about the research methodology and data analysis. VTE, venous thromboembolism; WES, whole-exome sequencing; MAF, minor allele frequency; VEP, variant effect predictor; QTL, quantitative trait locus. * Allele frequency obtained from gnomAD3 and 1000G databases. The hatched boxes contain the main study output. The hatched arrow indicates the inclusion of two prioritized frequent and functional variants in the STRING analysis. In bold, key terms of the study.

**Figure 2 ijms-24-13809-f002:**
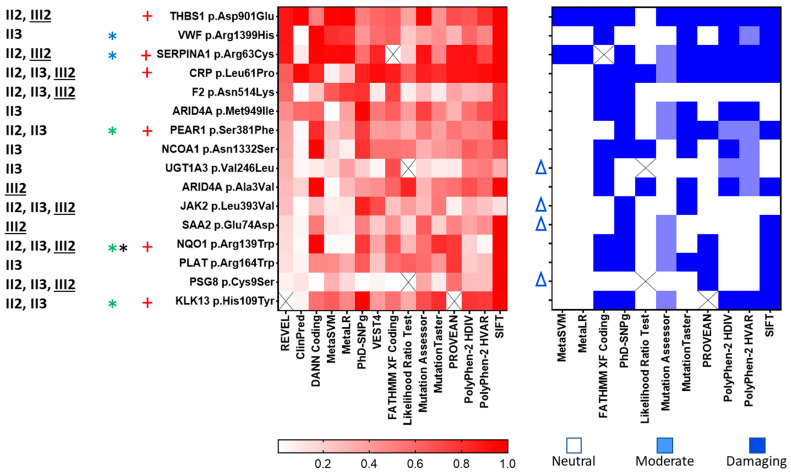
Heatmaps representing the degree of deleteriousness of variants. Full protein names are reported in the legend of Table 1. The 15 multi-component bioinformatics tools used for prediction, six of which exploiting artificial intelligence (ClinPred, DANN Coding, MetaSVM, REVEL, VEST4 and FATHMM XF), are reported. Variants are listed according to REVEL scores. The probability scores are reported in the left panel, according to the color scale below the panel. The right panel reports prediction scores, categorized as neutral, moderate or damaging by cut-off values within each tool. Affected family members, propositus III2 underlined, carrying the selected variants are listed on the left. +, variant detected in ≥2 thrombotic subjects and classified as «damaging» by ≥4 bioinformatics tools. *, variant associated with eQTL (green), sQTL (black), annotated in ClinVar (blue) and classified as «damaging» by ≥4 bioinformatics tools. Δ, variant non included in the interaction analysis (STRING).

**Figure 3 ijms-24-13809-f003:**
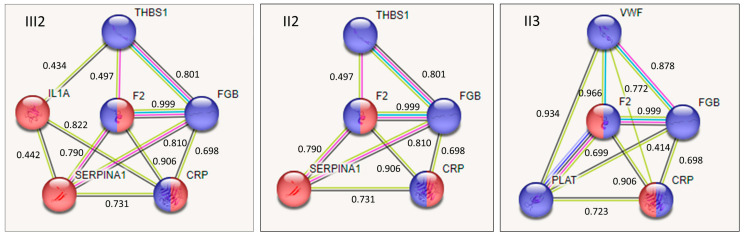
Functional interactions among proteins encoded by genes carrying the prioritized variants in each affected family member. Interactions predicted by STRING database (version 11.5). The protein nodes are reported with the corresponding gene symbol and filled with some known/predicted 3D structure. The proteins involved in the functional interactions are F2, prothrombin; SERPINA1, alpha-1 antitrypsin; CRP, C-reactive protein; THBS1, thrombospondin 1; PLAT, tissue plasminogen activator; VWF, von Willebrand factor; FGB, fibrinogen beta; IL1A, interleukin 1 alpha. The colored lines represent the different types of evidence used in predicting the associations (evidence mode); green line, neighborhood evidence; red line, gene fusion evidence; blue line, cooccurrence evidence; purple line, experimental evidence; light blue line, database evidence; black line, coexpression evidence. The combined score of the predicted interactions is indicated between each couple of proteins. Two significantly enriched terms are reported in the nodes: blue, proteins involved in the compartment «fibrinogen complex»; red, proteins involved in the “acute phase response”. Compartment «fibrinogen complex», FDR = 4.88e-07 (III2, II2) and 2.27e-09 (II3), and UniProt Keyword «Acute phase»/GO term “acute phase response”, FDR = 6.79e-05/5.09e-05 (III2), 6.79e-05/0.006 (II2) and 0.0097/not significant (II3).

**Figure 4 ijms-24-13809-f004:**
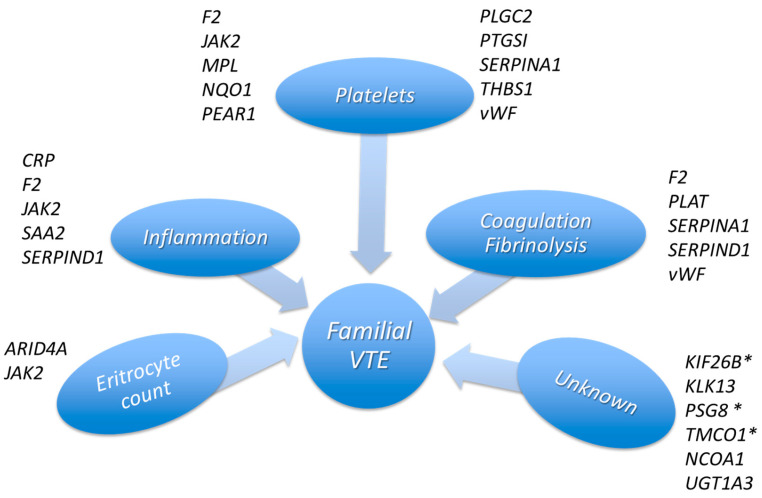
Genes, carrying low-frequency variants present in at least one affected family member, grouped by biological links. Genes are grouped according to biological processes or traits known to be linked to VTE. * Reported in Herrera-Rivero et al. [27] as involved in vascular repair (*KIF26B*) and platelet activation (*PSG8* and *TMCO1*).

**Table 1 ijms-24-13809-t001:** Variants (MAF < 0.04) present in at least one affected family member.

Gene Symbol	rsID dbSNP	cDNA	Protein Change	Frequency	Clinical Significance	Family Carriers
Non synonymous						
* **ARID4A** *	rs146509016	c.8C > T	p.Ala3Val	4.18819 × 10^−5^	NR	II1, III1, **III2**
* **ARID4A** *	rs1051029502	c. 2847G > C	p.Met949Ile	6.97876 × 10^−6^	NR	I1, **II3**
* **CRP** *	rs1376711485	c.182T > C	Leu61Pro	6.98295 × 10^−6^	NR	I1, **II2**, **II3**, III1, **III2**
* **F2** *	rs199772906	c.1542C > A	p.Asn514Lys	0.000244325	NR	I1, **II2**, **II3**, **III2**
* **JAK2** *	rs2230723	c.1177C > G	p.Leu393Val	0.0134624	B	I1, **II2**, **II3**, **III2**
* **KLK13** *	rs34089525	c.325C > T	p.His109Tyr	0.0217616	NR	**II2**, **II3**, III1
* **NCOA1** *	rs150066931	c.3995A > G	p.Asn1332Ser	0.0015095	NR	I1, **II3**
* **NQO1** *	rs1131341	c.415C > T	p.Arg139Trp	0.0255606	NR	I1, **II2**, **II3**, **III2**
* **PEAR1** *	rs77795865	c.1142 C > T	p.Ser381Phe	0.0230726	NR	**II2**, **II3**
* **PLAT** *	rs2020921	c.490 C > T	p.Arg164Trp	0.012879	NR	**II3**
* **PSG8** *	-	c.26G > C	p.Cys9Ser	-	-	I1, **II2**, **II3**, III1, **III2**
***SAA2***, ***SAA2-SAA4***	rs138605229	c.222A > C	p.Glu74Asp	0.00552043	NR	II1, III1, **III2**
* **SERPINA1** *	rs28931570	c.187C > T	p.Arg63Cys	0.00151399	P; LP	**II2**, III1, **III2**
* **THBS1** *	-	c.2703T > A	p.Asp901Glu	-	-	I1, **II2**, **III2**
* **UGT1A3** *	rs146461519	c.736G > T	p.Val246Leu	0.00043999	NR	**II3**
* **VWF** *	rs1800382	c.4196G > A	p.Arg1399His	0.00895051	B; LB; LP; P	**II3**
Synonymous						
* **KIF26B** *	rs201717788	c.507C > T	p.Val169=	0.00295131	NR	II1, III1, **III2**
* **MPL** *	rs544064034	c.1242G > A	p.Ser414=	6.98227 × 10^−6^	LB	**II2**, **II3**, **III2**
* **PLCG2** *	rs138637229	c.1146T > C	p.Phe382=	0.0071625	B; LB	**II2**
* **PTGIS** *	rs61322884	c.531C > T	p.Tyr177=	0.0221135	NR	**II3**
* **SERPIND1** *	rs35646566	c.423G > A	p.Leu141=	0.0173203	NR	I1, **II3**
* **TMCO1** *	rs78363884	c.486C > T	p.Leu162=	0.033832	B	**II2**

The gene symbols are reported according to NCBI Gene database (https://www.ncbi.nlm.nih.gov, accessed on 28 April 2023). Gene names: *ARID4A*, AT-rich interaction domain 4; *CRP*, C-reactive protein; *F2*, coagulation factor II (prothrombin); *JAK2*, Janus kinase 2; *KIF26B*, kinesin family member 26B; *KLK13,* kallikrein-related peptidase 13; *MPL*, proto-oncogene, thrombopoietin receptor; *NCOA1*, nuclear receptor coactivator 1; *NQO1*, NAD(P)H quinone dehydrogenase 1; *PEAR1*, platelet endothelial aggregation receptor 1; *PLAT*, tissue plasminogen activator; *PLCG2*, phospholipase C gamma 2; *PSG8*, pregnancy specific beta 1-glycoprotein 8; *PTGIS*, prostaglandin I2 synthase; *SAA2, SAA2-SAA4*, serum amyloid A2, serum amyloid A4; *SERPINA1*, alpha-1 antitrypsin; *SERPIND1*, serpin family D member 1 (heparin cofactor II); *THBS1*, thrombospondin 1; *TMCO1*, transmembrane and coiled coil domains1; *UGT1A3*, UDP glucuronosyltransferase family 1 member A3; *VWF*, von Willebrand factor. Allele frequency as reported in gnomAD3 all; Clinical significance, variant’s impact on disease annotated in ClinVar (NCBI resource; accessed April 2023); NR, not reported in ClinVar; B, benign; LB, likely benign; LP, likely pathological; P, pathological; **-**, no data in public databases. Family members affected by thrombosis are reported in bold.

**Table 2 ijms-24-13809-t002:** Functional impact of variants on (m)RNA expression.

Gene	rsID_dbSNP	Variant Region Features	Splicing Process *	QTL	
				eQTL	sQTL
* **ARID4A** *	rs146509016	missense; splice region #	ESE disruption	no data	no data
* **ARID4A** *	rs1051029502	missense	ESE disruption	no data	no data
* **F2** *	rs199772906	missense	ESE disruption	no data	no data
* **JAK2** *	rs2230723	missense	new ESS; ESE disruption; new donor	no data	no data
* **KLK13** *	rs34089525	missense	new ESS	*IGLON5*	not found
* **MPL** *	rs544064034	synonymous; enhancer	no predicted effect	no data	no data
* **NCOA1** *	rs150066931	missense	ESE disruption	no data	no data
* **NQO1** *	rs1131341	missense; splice region # open chromatin	donor site disruption	*NOB1; COG4;* *PDXDC2P*	*NQO1; NOB1; NPIPB14P*
* **PEAR1** *	rs77795865	missense	no predicted effect	*LRRC71*	not found
* **PLAT** *	rs2020921	missense	new ESE, new donor	*PLAT; POLB; AP3M2*	*SLC20A2; POLB*
* **SAA2-SAA4** *	rs138605229	missense	ESE disruption	ns	not found
* **UGT1A3** *	rs146461519	missense	new donor	no data	no data
* **PLGC2** *	rs138637229	synonymous ; CTCF site	new ESS	ns	not found
* **PTGIS** *	rs61322884	synonymous	ESE disruption	*SLC9A8*	not found
* **SERPIND1** *	rs35646566	synonymous; enhancer	ESE disruption	*AC000089.3*	not found
* **TMCO1** *	rs78363884	synonymous	new ESS	*RP11-466F5.10*	not found

The overlapped regulatory features were obtained from VEP, Variant Effect Predictor (Ensembl GRCh38 release 109, February 2023) and were referred to the MANE select transcripts; *, impact on splicing predicted from bioinformatics tools; # variant within the region of the splice site (1–3 bases in the exon or 3–8 bases in the intron); CTCF (CCCTC-binding factor), transcription factor; ESE, exonic splicing enhancer; ESS, exonic splicing silencer; new donor, new splice site donor. QTL, quantitative trait locus, taken from GTEx portal (Released (V8), accessed April 2023); eQTL, expression QTL (mRNA levels); sQTL: splicing QTL; no data, not investigated in GTEx portal; not found, no QTL found in GTEx portal; ns, no significant QTL reported in GTEx portal. *IGLON5*, IgLON Family member 5; *NOB1*, RNA-binding protein NOB1; *COG4*, conserved oligomeric Golgi complex subunit 4; *PDXDC2P*, pyridoxal-dependent decarboxylase domain-containing protein 2, pseudogene; *LRRC71*, leucine-rich repeat-containing protein 71; *POLB*, DNA polymerase beta; *AP3M2*, AP-3 complex subunit mu-2; *SLC20A2,* sodium dependent phosphate transporter 2; *SLC9A8*, sodium/hydrogen exchanger 8; *AC000089.3*, ribosomal protein L7a pseudogene; *RP11-466F5.10*, non-coding gene.

**Table 3 ijms-24-13809-t003:** WES studies in families with unexplained tendency for VTE.

Reference/*Strategy*	Population	Gene	rsID_dbSNP	MAF	Variant
**Cunha MLR et al., 2017** [34] *- Candidate genes (n = 126)* *- Variant MAF < 5%*	Dutch 2 Families n_a_ = 5 + 5	*STX2*	rs137928907	0.014	Phe32Val
		*ITGB3*	rs5918	0.121	Leu59Pro
		*APOH*	rs4581	0.353	Val266Leu
		*KLK8*	rs16988799	0.046	Val154Ile
		*KLK11*	rs3745539	0.063	Gly17Glu
**Chang WA et al., 2018** [32] *- Variant MAF < 1%* *- ClinVar annotation*	Asian n_a_ = 3	*SLC4A1*	rs121912749	0.000135	Gly130Arg
		*GP1BA*	rs770089708	0.109	Ser441fs
**Mulder R et al., 2020** [31] *- GWLA—Pathogenicity prediction* *- Rec. proteins assays—Molec. dynamics*	Dutch n_a_ = 5	*F2*	rs886048338	0.000004	Arg541Trp
**Morange PE et al., 2021** [33] *- MAF < 0.1%—Deleteriousness prediction-RNA seq in siRNAs targeted EC*	French n_a_ = 4	*MAST2*	rs1387081220	0.000004	Arg89Gln
**Present study** *- Candidate genes (n = 192)* *- MAF < 4%—Deleteriousness prediction—QTL analysis* *- Protein–protein interaction analysis*	Italian n_a_ = 3	*THBS1*	--	--	Asp901Glu
		*VWF*	rs1800382	0.008950	Arg1399His
		*SERPINA1*	rs28931570	0.001513	Arg63Cys
		*CRP*	rs1376711485	0.000007	Leu61Pro
		*F2*	rs199772906	0.000244	Asn514Lys
		*PLAT*	rs2020921	0.013	Arg164Trp

Strategies combined with WES are reported. GWLA, Genome-wide linkage analysis; Rec., recombinant; Molec., molecular; EC, endothelial cell. n_a_ = number of affected family members; -- no ID number. MAF, gnomAD-Exomes Global.

## Data Availability

All relevant data are included in the manuscript. Further original data will be made available by contacting the corresponding author within the regulations of the ethical approval.

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
