# Peer review of "Whole-Exome Sequencing in a Family with an Unexplained Tendency for Venous Thromboembolism: Multicomponent Prediction of Low-Frequency Variant Deleteriousness and of Individual Protein Interaction"

_ijms, 2023, doi:10.3390/ijms241813809_

Round 1

Reviewer 1 Report

The authors stated that “the family selected for Whole Exome Sequencing (WES) was chosen based on multiple findings supporting deep genetic analysis” However, the specific criteria used for selection were not explained in the method section. Moreover, pertinent information regarding the family members' ages, risk factor status, and potential underlying conditions, such as concomitant infections or kidney disorders that might have triggered the VTE, was absent from the method section.

Furthermore, the method section needed to provide details on whether there were additional family members who developed the disease or if they were unavailable for responding to questionnaires, which could impact the study's comprehensiveness and validity.

The lack of a flow chart in the study made the progression of information confusing, and its inclusion would have provided a clearer understanding of the research methodology and data analysis.

To gain further insights into the functional implications of the identified variants, it would be valuable for the authors to investigate if any of these variants result in protein functional defects. The authors could also measure the platelet and plasma levels of these proteins.

Additionally, performing platelet function studies and correlating them with the gene variants could offer crucial information about the genetic influence on platelet function and its association with VTE susceptibility.

Reviewer 2 Report

The authors present a family-based WES analyses combined with multiple bioinformatics approaches with focus on low frequency missense variants in a wide group of candidate genes.

The results indicate that WES has the potential to detect the most plausible combinations of low-frequency and small-size-effects variants, supporting venous thromboembolism susceptibility.

The identification of new candidate genes as well as a clear understanding of their role in the pathophysiology of VTE, are crucial to achieve a better identification of patients at higher risk.

The approach used by the authors to explain thrombosis in one family is comprehensive, combining WES and AI - based bioinformatics tools. In my opinion, it  provides advancement in the understanding of molecular processes in VTE.

The authors point out the several limitations of the study with acceptable explanations.

I have no major or minor concerns about the manuscript.

Round 2

Reviewer 1 Report

The authors' revisions have addressed all of the concerns raised by me. The manuscript is now ready for publication.